# Baicalin-Copper Complex Modulates Gut Microbiota, Inflammatory Responses, and Hormone Secretion in DON-Challenged Piglets

**DOI:** 10.3390/ani10091535

**Published:** 2020-08-31

**Authors:** Andong Zha, Zhijuan Cui, Ming Qi, Simeng Liao, Jia Yin, Bie Tan, Peng Liao

**Affiliations:** 1Laboratory of Animal Nutritional Physiology and Metabolic Process, Key Laboratory of Agro-Ecological Processes in Subtropical Region, National Engineering Laboratory for Pollution Control and Waste Utilization in Livestock and Poultry Production, Institute of Subtropical Agriculture, Chinese Academy of Sciences, Changsha 410125, Hunan, China; anton1223@163.com (A.Z.); czj123@stu.hunau.edu.cn (Z.C.); qiming16@mails.ucas.ac.cn (M.Q.); liaosaimeng16@mails.ucas.ac.cn (S.L.); bietan@isa.ac.cn (B.T.); 2University of Chinese Academy of Sciences, Beijing 100008, China; 3College of Animal Science and Technology, Hunan Agriculture University, Changsha 410128, Hunan, China; 4Hunan Provincial Key Laboratory of Animal Function and Regulation, College of Life Sciences, Hunan Normal University, Changsha 410081, Hunan, China; jiayin@hunnu.edu.cn

**Keywords:** baicalin-copper complex, deoxynivalenol, piglets, hormone, intestinal microbiota, inflammatory cytokines

## Abstract

**Simple Summary:**

Deoxynivalenol (DON) is the most common mycotoxin contaminant in the agriculture industry worldwide. Copper is very efficacious in promoting growth performance and improving feed remuneration, and baicalin may alleviate oxidative stress and inflammatory responses in humans and animals. We speculated that the combined effect of baicalin and copper would have some effect in DON-challenged piglets. The present study examined the effects of a baicalin-copper complex on inflammatory responses, hormone secretion, and gut microbiota in DON challenged piglets. These findings provide new application prospects in piglets involving the combination of baicalin and copper.

**Abstract:**

The present experiment assessed the inflammatory responses, hormone secretion, and gut microbiota of weanling piglets administered baicalin-copper complex (BCU) or deoxynivalenol (DON) supplementation diets. Twenty-eight piglets were randomly assigned to four groups: control diet (Con group), a 4 mg DON/kg diet (DON group), a 5 g BCU/kg diet (BCU group), a 5 g BCU + 4 mg DON/kg diet (DBCU group). After 14 days, the results showed that dietary BCU supplementation remarkably increased the relative abundance of *Clostrium bornimense* and decreased the relative abundance of *Lactobacillus* in the DBCU group (*p* < 0.05). BCU decreased the serum concentration of IgG, IL-2, IFN-γ, and IgA in DON treated piglets (*p* < 0.05), and promoted the serum concentration of IL-1β, IgG, IL-2, IFN-γ, IgA, IL-6, IgM, and TNFα in normal piglets (*p* < 0.05). BCU increased the concentrations of serum IGF1, insulin, NPY, GLP-1, and GH, and decreased the concentrations of serum somatostatin in no DON treated piglets (*p* < 0.05). Dietary BCU supplementation significantly promoted the secretion of somatostatin, and inhibited the secretion of leptin in piglets challenged with DON (*p* < 0.05). BCU regulated the expression of food intake-related genes in the hypothalamus and pituitary of piglets. Collectively, dietary BCU supplementation alleviated inflammatory responses and regulated the secretion of appetite-regulating hormones and growth-axis hormones in DON challenged piglets, which was closely linked to changes of intestinal microbes.

## 1. Introduction

The mycotoxin deoxynivalenol (DON) is primarily produced by *Fusarium* spp., and it is commonly found in cereals [1,2,3]. DON is the most common mycotoxin contaminant in agriculture industry worldwide [4]. Christiane et al. analyzed 74,821 feed and feed ingredient samples from 100 countries and found that the detection rate of DON was as high as 64% [5]. Low-dose DON ingestion in humans and animals causes growth retardation, metabolic disorders, and inflammatory responses, and high-dose DON exposure leads to vomiting, diarrhea, and gastrointestinal bleeding [6,7,8,9]. Previous research showed that DON induced growth retardation may be related to interference with hormone secretion and immune responses [10,11,12,13]. Some studies reported that DON destroyed the balance of gut microbiota to trigger intestinal inflammation [14,15,16]. Therefore, it is necessary to maintain the intestinal microbiota balance, ameliorate inflammatory responses, and regulate hormone secretion to promote the growth performance of DON challenged piglets. 

The root of *Scutellariae baicalensis* Georgi is used to treat some diseases, such as hypertension, bacterial and viral infections, and inflammation, and baicalin is one of the main bioactive components of this root. Previous studies reported that baicalin downregulated the concentration of several cytokines and proteins, such as tumor necrosis factor-α (TNF-α), immunoglobulin E (IgE), interleukin-6 (IL-6), and interleukin-1β (IL-1β), to ameliorate inflammatory disorders and tissue injury [17,18,19]. Baicalin also effectively reduced oxidative stress and apoptosis via the Nrf2 pathway [20,21]. Abundant scientific evidence also shows that baicalin promoted neuronal protective factors expression and neurogenesis, which were related to the anti-oxidant and anti-inflammatory activities of baicalin [22,23]. A few studies reported that the intestinal microbiota influenced the absorption of baicalin [24]. Copper is an essential mineral for a variety of functions in humans and animals ranging from tissue differentiation to immune responses. Copper is used to improve the survival rate and promote growth in weanling piglets. Dietary copper supplementation in pig production promoted growth performance and improved feed remuneration. A baicalin-copper complex (BCU) is a complex of baicalin and copper. Compared to baicalin alone, BCU exhibited stronger anti-bacterial and anticancer effects [25,26]. Therefore, BCU is an important consideration as an alternative feed ingredient in weanling piglets.

Therefore, we hypothesized that BCU supplementation would affect hormone secretion, intestinal microbiota composition, and inflammatory responses in DON-challenged piglets.

## 2. Results

### 2.1. 16S rRNA Sequencing

We sequenced the V3 and V4 regions of the 16s rRNA gene from ileac digesta [27]. An average of 81,759 raw reads were obtained from samples, and 76,671 clean tags were obtained after quality control. The clean tags were clustered into OTUs (operational taxonomic units). A total of 1001 OTUs were generated, and the number of core OTUs was 316 (Figure 1C). According to the rarefaction curve, the number of OTUs that reached the plateau stage and no longer changed rapidly with the increase in sequencing sequences indicated that the 16s rRNA sequencing had basically detected all microorganisms in the ileum (Figure 1A). The rank abundance curve shows that the DON group had the most abundant microbial composition and the highest uniformity, followed by the DBCU group. The Con group and the BCU group had the lowest intestinal microbial richness (Figure 1B).

The overall microbial composition differed at different taxonomic levels. As the figure shows, the microbiota of the ileac digesta were dominated by *Firmicutes* (88.00%), *Proteobacteria* (7.96%), *Bacteroidetes* (2.11%) at the phylum level (Figure 1D). At the genus level, OTUs associated with unidentified *Clostridiales* (55.99%), *Romboutsia* (5.62%), and *Streptococcus* (4.99%) were predominant in the ileac digesta (Figure 1E).

### 2.2. Gut Microbiota Diversity

The observed species indicated the number of species contained in the sample. Notably, dietary BCU supplementation lead to an increase in observed species (Figure 2A). The figure shows that no differences were detected in diversity indices (Shannon), but there was an increase in richness estimators (Chao1 and Ace) in the dietary BCU supplementation groups (Figure 2B–D).

Based on the binary Jaccard distance, we applied principal coordinates analysis (PCoA) plots to evaluate the differences in beta-diversity. Dietary DON supplementation caused gradual and significant segregation changes in the intestinal microbiota. The Con group and the BCU group clustered together, which suggested that dietary BCU supplementation had less effect on the intestinal composition of normal piglets (Figure 2E). Consistently, nonmetric multidimensional scaling (NMDS) plots based on the binary Jaccard distance confirmed the effect of DON on the intestinal microbiota (Figure 2F). The figure shows that the microbial community in DON-supplementation piglets was more closely related to the DBCU group (Figure 2G), which suggested that DON had a greater impact on gut microbes than BCU.

### 2.3. LEfSe Analysis

Our data showed that experiment treatment caused microbiota changes. To identify different strains between the four groups, we performed linear discriminant analysis effect size (LEfSe) analysis from the sequencing data. Compared to the Con group, the DON group had a higher relative abundance of *Oceanobacillus* (Figure 3A). The relative abundances of *Neisseria*, *Desulfovibrionaceae*, *Negativibacillus*, *Alistipes*, *Lactobacillaceae*, and *Alloprevotella* in the Con group were much higher than the BCU group, and the relative abundances of unidentified *Enterobacteriaceae*, *Clostridum bornimense*, *Clostridium butyricum*, and *Clostridium disporicum* in the BCU group were much higher than the Con group (Figure 3C). *Moraxellaceae* and *Clostridium bomimense* were higher in the DBCU group than the DON group, and *Lactobacillus* was higher in the DON group than the DBCU group (Figure 3B).

### 2.4. Serum Immunoglobulins, Cytokines, and Biochemical Indexes

To explore the effects of dietary DON and BCU on the immune response, serum concentrations of immunoglobulins and inflammatory cytokines were measured after the experiment. As revealed in Figure 4, the interaction effect between DON and BCU on serum concentrations of immunoglobulins and inflammatory cytokine levels was statistically significant (*p* < 0.01), i.e., the effect of BCU on serum concentrations of immunoglobulins and inflammatory cytokine levels was different in different DON treatments. Simple effects analysis showed that BCU significantly promoted the serum concentrations of immunoglobulin A (IgA), immunoglobulin G (IgG), immunoglobulin M (IgM), interferon-gamma γ (IFN-γ), interleukin-1β (IL-1β), TNFα, and interleukin-6 (IL-6) in normal piglets (*p* < 0.01), and the serum concentrations of IgA, IgG, interleukin-2 (IL-2), and IFN-γ of the DBCU group increased significantly compared to the DON group (*p* < 0.05).

As revealed in Appendix A, the interaction effect between DON and BCU on serum biochemical indexes was not statistically significant (*p* > 0.05). The main effect analysis showed that the serum glucose (GLU) concentration in no BCU-treated piglets was 4.22 ± 0.19, and the serum GLU concentration in the BCU-treated piglets was 4.82 ± 0.19. Therefore, BCU treatment significantly increased the serum concentrations of GLU in piglets (*p* < 0.05). BCU treatment also had a tendency to change the serum concentrations of albumin (ALB), blood urea nitrogen (BUN), and total cholesterol (CHOL) in piglets (*p* = 0.07, *p* = 0.05, *p* = 0.07).

### 2.5. Hormone Concentrations in Serum

DON causes growth retardation, which may be related to hormone secretion. Thus, we analyzed serum growth axis-associated hormones and feed intake-related hormones. As shown in the histogram (Figure 5), the interaction effect between DON and BCU significantly affected the concentrations of insulin-like growth factor 1 (IGF1), somatostatin (SS), insulin, and growth hormone (GH) in piglets (*p* < 0.01). Compared to the Con group, the BCU group exhibited significantly increased concentrations of IGF1, insulin, and GH in serum, and significantly decreased concentrations of SS in serum (*p* < 0.05). Compared to the DON group, the concentration of SS in the DBCU group was significantly increased. These results show that BCU supplementation had little effect on serum feed intake-related hormones (Figure 3). The interaction effect between DON and BCU significantly affected the concentrations of leptin, GLP1, and neuropeptide Y (NPY) in piglets (*p* < 0.05). Simple effects analysis showed that BCU supplementation significantly promoted the secretion of NPY and GLP-1 in no DON induced piglets (*p* < 0.05), and it significantly inhibited the secretion of leptin in DON treated piglets (*p* < 0.05).

### 2.6. Relative mRNA Expression of Hypothalamus and Pituitary Genes

We detected the expression of six genes affected by dietary DON or BCU supplementation in the hypothalamus (Figure 6A). Specifically, dietary BCU significantly influenced (*p* < 0.05) the expression of somatostatin (SS), and HTR3A1, and dietary DON significantly influenced (*p* < 0.05) the expression of INR, AKT, and HTR3A2. Notably, dietary BCU supplementation upregulated the mRNA levels of SS and HTR3A1 (*p* < 0.05).

The expression of several hormone and receptor genes in the pituitary were tested using real-time q-PCR (Figure 6B). The mRNA levels of INR, CCK-1R, AKT, and COX-2 in the pituitary were significantly affected by the interaction effect between DON and BCU. Specifically, dietary BCU supplementation upregulated the mRNA expression of INR, CCK-1R, and COX-2 in no DON treated piglets, along with downregulated the mRNA expression of AKT in DON-treated piglets (*p* < 0.05). BCU supplementation obviously influenced the mRNA expression of GLP2 and 5-HT in the pituitary (*p* < 0.05), and DON supplementation obviously affected the mRNA levels of AGRP, NPY, and POMC in the pituitary (*p* < 0.05). Notably, dietary BCU upregulated the mRNA expression of GLP2 (*p* < 0.05) and downregulated the mRNA levels of 5-HT (*p* < 0.05). Similarly, dietary DON supplementation upregulated the expression of AGRP and POMC (*p* < 0.05) and downregulated the mRNA levels of NPY (*p* < 0.05).

## 3. Discussion

DON induces anorexia, growth retardation, and inflammatory responses in human and animals [1,3]. Abundant scientific evidence shows that DON leads to changes in the composition of the intestinal microbiota, stimulates the immune system, and induces the secretion of gut hormones, which act on the pituitary and hypothalamus via the brain–gut axis, resulting in decreased appetite and growth retardation in humans and animals [28]. The present study described the impacts of dietary BCU supplementation on the intestinal microbiota composition, inflammatory responses, and hormone secretion in DON-challenged piglets.

DON induces activation of NF-*k*B, which triggers inflammatory responses and selectively induces upregulation of a range of cellular cytokines, chemokines, and other immune-related inflammatory factors [12,29]. Our experimental results showed that DON significantly increased the concentration of immunoglobulins and inflammatory cytokines, which indicates that DON leads to inflammatory response in piglets. Baicalin was well studied for its beneficial role in alleviating oxidative stress and inflammatory responses in humans and animals [18,30]. The present study also showed that the levels of IL-2, IgA, IFN-γ, and IgG in the DBCU group were significantly lower than the DON group. This result indicated that BCU suppressed inflammatory responses in DON-challenged piglets. Notably, BCU promoted inflammatory responses in normal piglets, which may be due to the high concentration of copper.

GH and IGF-1 play a pivotal part in regulating the growth and development of livestock. Amuzie et al. showed that the toxicological effects of DON lead to growth retardation via suppression of the secretion of growth axis hormone [31]. Our results showed that the levels of GH and IGF-1 in the DON group were remarkably lower than the Con group, which is consistent with a previous study that showed that the growth performance of the DON group was also worse than the Con group [32]. The BCU group in the present study exhibited significantly increased concentrations of IGF1, insulin, and GH in serum compared to the Con group. BCU also remarkably decreased the concentrations of serum SS. These findings indicate that BCU may affect the secretion of hormones to improve the growth performance of piglets. However, the secretion of SS in the DBCU group was significantly higher than the DON group, which indicates that BCU may have different effects on growth under normal and pathological conditions.

DON crosses the blood–brain barrier and targets the hypothalamus to produce anorexia behavior, and it affects the secretion of gut hormones, such as CCK, PYY, GLP-1, and 5-HT, which target the hypothalamus via the brain-gut axis to regulate appetite [33,34,35]. DON significantly promoted the levels of GLP1, leptin, and insulin compared to the Con group in the present study, and GLP1 and leptin lead to reduced food intake [36,37]. 5-HT plays an important role in the DON-induced suppression of feed intake [28,38]. BCU downregulated the expression of HTR3A1 in the hypothalamus in no DON-treated piglets in the current study, and downregulated the expression of 5-HT in the hypothalamus in DON-treated piglets, which shows that BCU promoted piglets to eat normal feed and inhibited piglets from eating DON-contaminated feed. Dietary BCU supplementation also suppressed the expression of 5-HT in the pituitary. Previous studies suggested that NPY was an important orexigenic peptide, and the orexigenic effect of NPY was associated with AGRP and POMC [33,39]. Dietary DON supplementation promoted the expression of AGRP and POMC, and suppressed the expression of NPY in the pituitary. In conclusion, BCU may promote feed intake via regulation of hormone secretion, and DON inhibits food intake by regulating hormones secretion.

Intestinal microbes play a pivotal part in host health by regulating immunity and inhibiting pathogenic bacteria [40]. Dietary BCU or DON are important factors that affected the intestinal microbiota in animals [15,24]. We used Illumina HiSeq sequencing, and determined the microbiota composition in the ileal digesta of the four groups. Our results are consistent with previous studies that the microbiota of swine were dominated by *Firmicutes*, *Proteobacteria*, and *Bacteroidetes* (Figure 1D–E) [41]. BCU addition significantly increased the observed species, Chao 1, and Ace compared to the no BCU treatment piglets (Figure 2A–D). A decrease microbial diversity is generally linked with inflammation [42]. The increased relative abundance of *Lactobacillus* is associated with weight loss [43]. On the genus level, we observed that the relative abundance of *Lactobacillus* was significantly higher in the DON group than the DBCU group, and the relative abundance of *Lactobacillus* was dramatically higher in the Con group than the BCU group. Some studies also reported that most of the *Lactobacillus* assayed induced inflammatory responses, which suggests that DON promoted the release of inflammatory factors by changing the abundance of *Lactobacillus* in the intestinal microbiota [44,45]. The expression of GHR and IGF1 in the liver of broilers fed with *Lactobacillus *plantarum** strains RI11 increased significantly, and these findings indicate that DON and BCU may regulate growth by changing the abundance of *Lactobacillus* [46]. The relative abundance of *Clostridium butyricum* was significantly lower in the Con group than the BCU group. *Clostridium butyricum* is widely used as a feed additive to enhance immune responses and promote growth in swine production [47,48,49]. These results indicated that DON and BCU may change the intestinal composition, such as *Lactobacillus* and *Clostridium butyricum*, to regulate inflammatory responses and weight gain in piglets. However, this hypothesis must be verified in sterile animal models.

## 4. Materials and Methods

### 4.1. Preparation of the Baicalin-Copper Complex

An equimolar amount of baicalin was added to a 1% sodium bicarbonate aqueous solution and stirred to completely dissolve it. Copper sulfate was added according to the molar ratio of baicalin to copper sulfate of 1:2, and after stirred for 4 h at room temperature. A yellow precipitate was obtained via filtration. The yellow precipitate was washed with water and dried to obtain the baicalin-copper complex.

### 4.2. Diets, Animals, and Sample Collections

The preparation of DON contaminated feed was prepared according to previous studies, and the doses of DON and BCU were similar to Zha et al. [21,25,32]. The DON contaminated feed was diluted with high-concentration DON-contaminated feed, and BCU was added to the feed in proportion. Piglets (n = 28, Landrace × Yorkshire, 28 d age), with an average initial body weight of 6.15 ± 0.13 kg, were allocated randomly to four groups. Piglets received one of four experimental diets based on the corn and soybean meal included the control diet (Con), a 4 mg DON/kg diet (DON), a 5 g BCU/kg diet (BCU), a 5 g BCU + 4 mg DON/kg diet (DBCU) [50]. The composition of the control diet is presented in Zha et al. [23]. During the 14-day trial, piglets were housed individually in stalls with plastic slatted flooring (one pig per pen), and the room temperature was maintained at approximately 30 °C. Water and feed were provided ad libitum. During the trial, one piglet in the Con group died. On 15 d, seven piglets were randomly selected from each group for slaughter, and serum, the hypothalamus, pituitary, and intestinal chyme samples were collected, and stored at −80 °C. The experiments were supervised by the Animal Care and Use Committee of the Institute of Subtropical Agriculture (approval code: ISACAS protocol #20180326, approval date: 26 March 2018) [21,51,52].

### 4.3. Gut Microbiota Analysis

Gut microbiota analyses were performed by a commercial company (Novogene, Beijing, China).

#### 4.3.1. Microbial Genomic DNA Extraction and PCR Amplification

Microbial genomic DNA from samples was extracted using a CTAB/SDS assay kit and diluted to 1 ng/μL using sterile water. The V3-V4 region of the bacteria 16S ribosomal RNA gene was amplified using primers (515F (5′-GTGCCAGCMGCCGCGGTAA-3′) and 806R (5′-GGACTACHVG GGTWTCTAAT-3′). PCR reactions were performed in a 30 μL mixture containing 15 μL of Phusion^®^ High-Fidelity PCR Master Mix (New England Biolabs); 2 μM of each primer, and approximately 10 ng template DNA. The following PCR protocol was used: (1) initial denaturation (98 °C/1 min); (2) amplification (98 °C/10 s, 50 °C/30 s, 72 °C/30 s, 30 cycles); (3) final extension (72 °C/5 min).

#### 4.3.2. 16S rRNA Sequencing

After mixing PCR products in equidensity ratios, we purified the products using the Gene JET Gel Extraction Kit (Thermo Scientific, Waltham, MA, USA).

Sequencing libraries were generated using the NEB Next^®^ Ultra™ DNA Library Prep Kit for Illumina (NEB, Beverly, MA, USA). After quantification and assessment with a Qubit@ 2.0 Fluorometer (Thermo Scientific) and Agilent Bioanalyzer 2100 system, the libraries were paired-end sequenced (2 × 250) on an Illumina HiSeq platform according to standard protocols.

#### 4.3.3. Bioinformatics Analysis

Raw data were processed using Cutadapt (V1.9.1) and Usearch to get clean data [53,54,55]. Based on clean data, the UPARSE (Uparse v7.0.1001) was performed to cluster the operational taxonomic unit (OTU) with ≥ 97% sequence similarity [56]. The taxonomy of each OTU was analyzed using Mothur against the SILVA132 (SSUrRNA, http://www.arb-silva.de/) database (threshold 0.8–1) [57,58]. Alpha diversity, including the observed species, Chao1, abundance-based coverage estimator (ACE), Simpson, and Shannon diversity indices, were calculated using QIIME (Version 1.9.1) [59]. Rarefaction curves, and rank abundance curves were drawn using software R (Version 2.15.3) [60]. For beta-diversity analysis, principal coordinate analysis (PCoA) [61], principal component analysis (PCA) [62], nonmetric multidimensional scaling (NMDS) [63], and unweighted pair group method with arithmetic mean (UPGMA) were performed using QIIME (Version 1.9.1) and R (Version 2.15.3) [64,65]. The bacterial differences between groups were assessed using LEfSe analysis (LDA score = 3.5).

### 4.4. Serum Biochemical Index, Immunoglobulins, and Cytokines

Serum activities of blood urea nitrogen (BUN), albumin (ALB), glucose (GLU), and total cholesterol (CHOL) were analyzed using a CX-4 Automatic Biochemical Analyzer (Beckman, USA) and commercial kits according to the manufacturers’ instructions [66,67].

Concentrations of serum immunoglobulin A (IgA), tumor necrosis factor-α (TNFα) immunoglobulin G (IgG), interleukin-6 (IL-6), immunoglobulin M (IgM), interferon-gamma (IFN-γ), and interleukin-1β (IL-1β) were measured using commercial ELISA kits (Nanjing Jiancheng Bioengineering Institute, Nanjing, China). The plates were read at 450 nm and the concentrations of the samples were determined from a standard curve [68].

### 4.5. Determination of Serum Hormone

Concentrations of serum insulin-like growth factor 1 (IGF-1), somatostatin (SS), growth hormone (GH), peptide YY (PYY), proopiomelanocortin (PMOC), glucagon-like peptide-1 (GLP1), neuropeptide Y (NPY), 5-hydroxytryptamine (5-HT), agouti related protein (AGRP) were measured using commercial ELISA kits (Nanjing Jiancheng Bioengineering Institute, Nanjing, China). The plates were read at 450 nm and the concentrations of the samples were determined from a standard curve [68].

### 4.6. Expression of the Hypothalamus and Pineal Genes

Total RNA was extracted from the hypothalamus and pituitary from each group. Real-time q-PCR was performed as previously described [32,69]. Primers are presented in Appendix A. Glyceraldehyde phosphate dehydrogenase (GAPDH) and β-actin were used to normalize target gene transcript levels. The relative expression of genes was calculated using the formula 2^−ΔΔCt^.

### 4.7. Statistical Analysis

Two-way MANOVA was used to test the main effects of dietary DON and BCU supplementation, and their interactions. Missing values were filled in with means. When there were outliers, the data were logarithmically transformed before MANOVA. If there was a main effect or the interaction was significant, then Duncan’s test was used for post hoc comparisons of means. Data are presented as means ± SEMs. The significance level was set at *p* < 0.05. Statistical analyses were performed using SPSS Statistics software (version 20, IBM). Figures were prepared using GraphPad Prism 7.0 (GraphPad Software, La Jolla, CA, USA) [50,70,71].

## 5. Conclusions

In summary, dietary BCU supplementation alleviated inflammatory responses and regulated the secretion of appetite-regulating hormones and growth-axis hormones in DON-challenged piglets, and dietary BCU or DON supplementation to piglets changed the composition of the intestinal microbiota. The effects of DON and BCU on inflammation responses and hormone secretion may be closely linked to the changes in intestinal microbes. The present study provides new application prospects in piglets involving the combination of baicalin and copper.

## Figures and Tables

**Figure 1 animals-10-01535-f001:**
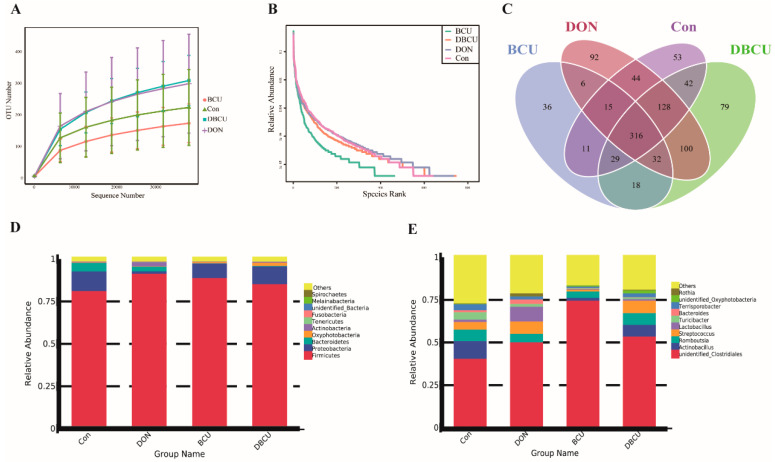
Sequencing quality analysis and the composition of intestinal microbiota. (**A**) Rarefaction curve; (**B**) rank abundance curve; (**C**) Venn diagrams for bacterial OTUs (operational taxonomic units); (**D**) taxonomic composition of the ileac digesta at the phylum level; (**E**) taxonomic composition of the ileac digesta at the genus level. Basal diet (Con); 4 mg deoxynivalenol (DON)/kg diet (DON); 5 g baicalin-copper complex (BCU)/kg diet (BCU); 5 g BCU/kg with 4 mg DON/kg diet (DBCU).

**Figure 2 animals-10-01535-f002:**
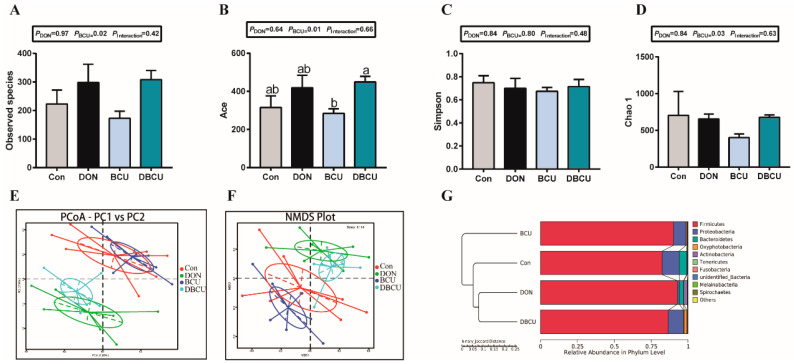
Effects of dietary DON and BCU on gut microbiota diversity in piglets. (**A**) Observed species; (**B**) Shannon H index; (**C**) Chao 1 index; (**D**) Ace index. (**E**) Principal coordinates analysis (PCoA) of intestinal microbiomes (binary Jaccard). (**F**) Nonmetric multidimensional scaling (NMDS) of intestinal microbiomes (binary Jaccard). (**G**) Comparison of binary Jaccard between pairs of samples. (**A**–**D**) Values are means ± SEMs, n = 7. Different letters in the picture represent differences, *p* < 0.05. Basal diet (Con); 4 mg DON/kg diet (DON); 5 g BCU/kg diet (BCU); 5 g BCU/kg with 4 mg DON/kg diet (DBCU).

**Figure 3 animals-10-01535-f003:**
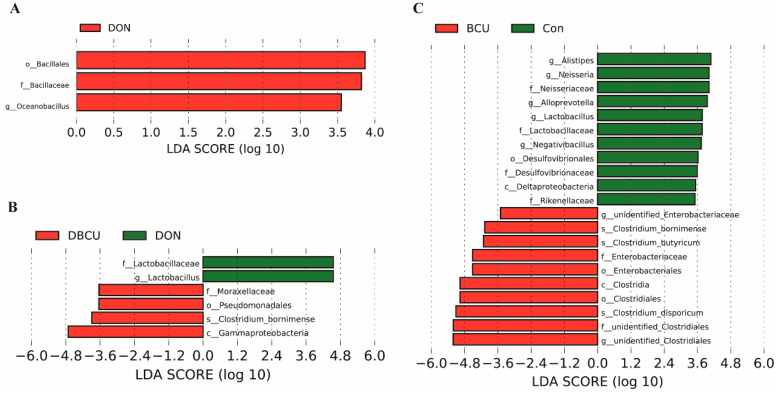
Linear discriminant analysis effect size (LEfSe) analysis of the piglet intestinal microbiota data. (**A**) DON and Con group, (**B**) DON and DBCU group, (**C**) BCU and Con group. LEfSe identified significantly different bacterial taxa (LDA Score>3.5). Basal diet (Con); 4 mg DON/kg diet (DON); 5 g BCU/kg diet (BCU); 5 g BCU/kg with 4 mg DON/kg diet (DBCU).

**Figure 4 animals-10-01535-f004:**
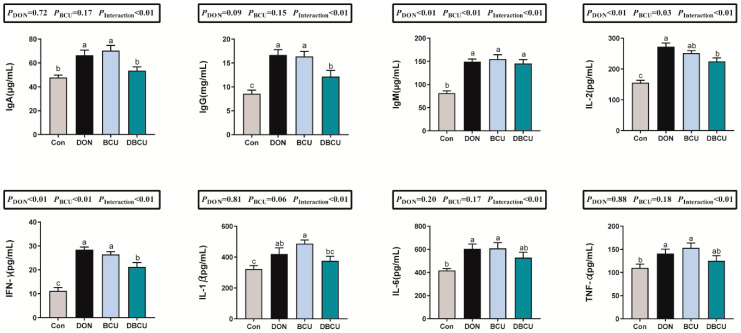
Effects of dietary DON and BCU on serum immunoglobulins, cytokines, and biochemical indexes in piglets. Values are means ± SEMs, n = 7. Different letters in the picture represent differences, *p* < 0.05. Basal diet (Con); 4 mg DON/kg diet (DON); 5 g BCU/kg diet (BCU); 5 g BCU/kg with 4 mg DON/kg diet (DBCU). IL-2: interleukin-2; IgA: immunoglobulin A; TNFα: tumor necrosis factor-α; IgM: immunoglobulin M; IFN-γ: interferon-gamma γ; IL-1β: interleukin-1β; IgG: immunoglobulin G; IL-6: interleukin-6; ALB: albumin; BUN: blood urea nitrogen; CHOL: total cholesterol; GLU: glucose.

**Figure 5 animals-10-01535-f005:**
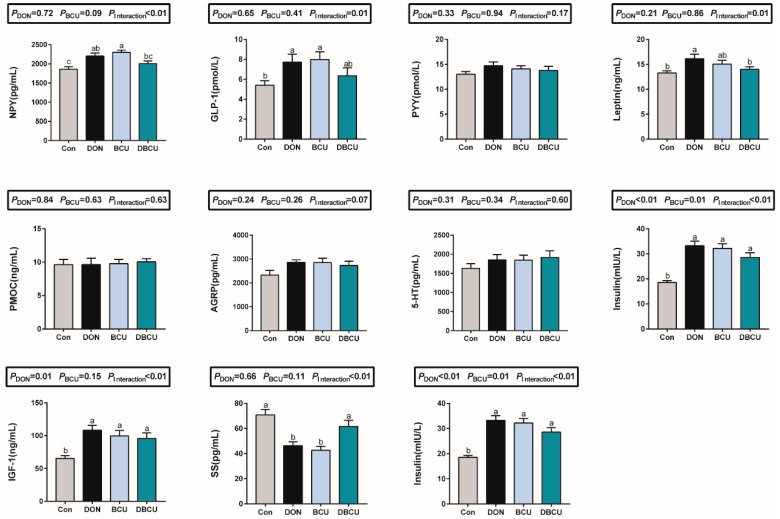
Effects of dietary DON and BCU on serum hormone in piglets. Values are means ± SEMs, n = 7. Different letters in the picture represent differences, *p* < 0.05. Basal diet (Con); 4 mg DON/kg diet (DON); 5 g BCU/kg diet (BCU); 5 g BCU/kg with 4 mg DON/kg diet (DBCU). IGF1: insulin-like growth factor 1; SS: somatostatin; GH: growth hormone; PYY: peptide YY; PMOC: proopiomelanocortin; GLP1: glucagon-like peptide-1; NPY: neuropeptide Y; AGRP: agouti related protein; 5-HT: 5-hydroxytryptamine.

**Figure 6 animals-10-01535-f006:**
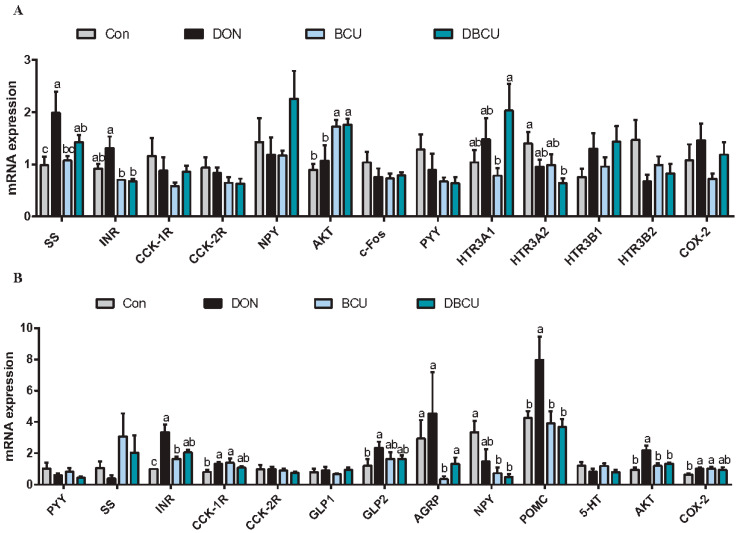
Effects of dietary DON and BCU on mRNA expression in the hypothalamus and pituitary of piglets. (**A**) Hypothalamus. (**B**) Pituitary. Values are means ± SEMs, n = 7. Different letters in the picture represent differences, *p* < 0.05. Basal diet (Con); 4 mg DON/kg diet (DON); 5 g BCU/kg diet (BCU); 5 g BCU/kg with 4 mg DON/kg diet (DBCU). PYY: peptide tyrosine; SS: somatostatin; c-Fos: Fos proto-oncogene, AP-1 transcription factor subunit; GLP-1R: glucagon like peptide 1 receptor; CCK-1R: cholecystokinin type A receptor; INR: insulin receptor; GLP-2R: glucagon like peptide 2 receptor; CCK-2R: cholecystokinin B receptor; AGRP: agouti-related protein; NPY: neuropeptide Y; COX-2: cyclooxygenase-2 (also known as prostaglandin-endoperoxide synthase 2 (PTGS2)); POMC: pro-opiomelanocortin; AKT: AKT serine/threonine kinase 1 (AKT1), transcript variant X3; HTR3A: 5-hydroxytryptamine receptor 3A; HTR3B: 5-hydroxytryptamine receptor 3B.

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
