# Peer review of "Baicalin-Copper Complex Modulates Gut Microbiota, Inflammatory Responses, and Hormone Secretion in DON-Challenged Piglets"

_animals, 2020, doi:10.3390/ani10091535_

Round 1

Reviewer 1 Report

The comments and suggestions for the Authors may be found below, please:

  • typo errors should be corrected: "georgi" (should be written using capitals and without using italics); "Dietary" (line 256); the names of the hormones do not need to be written using capitals;
  • "so on" (line 44) should be avoided due to a question of style. It is suggested using "among others" or "to cite a few";
  • please, check the subtitle 2.1.16 (line 60);
  • the quality of the images is excellent;
  • the Results may be described in a more direct/objective way (please, see the text between the lines 119-123 as an example;
  • the Discussion is succint and is not speculative.

Author Response

Dear. Editor,

   Thank you very much for your letter and advice. We have revised the manuscript, and would like to re-submit it for your consideration. Following your suggestions, the manuscript has been reviewed and modified by a native English speaker; the amendments are shown in red in the revised manuscript.

We have addressed the comments raised by the reviewers, and the amendments are also shown in red in the revised manuscript. Point-by-point responses to the reviewers’ comments are listed below this letter. Spelling and grammatical mistakes have been addressed by two native English-speaking editors from American Journal Experts.

We hope that the revised version of the manuscript is now acceptable for publication in your journal.

I look forward to hearing from you soon. 

With best wishes,

Yours sincerely,

Corresponding authors: Dr. Peng Liao

liaopeng@isa.ac.cn (Peng Liao); Tel/ Fax: +86-731-84619703  

We would like to express our sincere thanks to the reviewers for constructive and positive comments.

Response to Reviewer 1

typo errors should be corrected: "georgi" (should be written using capitals and without using italics);

Response: typo errors has been corrected according to your suggestion (line 59).

"Dietary" (line 256)

Response: Dietary has been added in line 281

the names of the hormones do not need to be written using capitals

Response: we have checked and revised the hormone names in the full text.

"so on" (line 44) should be avoided due to a question of style. It is suggested using "among others" or "to cite a few"

Response: We have deleted ‘so on’ in the article

please, check the subtitle 2.1.16 (line 60);

Response:it should be 2.1. 16S rRNA sequencing, we have add a space after ‘2.1.’

the Results may be described in a more direct/objective way (please, see the text between the lines 119-123 as an example;

We have modified some results based on your suggestions.

the Discussion is succint and is not speculative.

We have modified the discussion section as required.

Reviewer 2 Report

In the manuscript (Baicalin-Copper Complex Modulates Gut Microbiota, Inflammatory Responses, and Hormone Secretion in DON Challenged Piglets), the authors examined the effect of baicalin-copper complex or deoxynivalenol supplementation diets on inflammatory responses, hormone secretion, and gut microbiota of weanling piglets. The manuscript provides valuable information and well-designed study. However, there are some modifications needed.

-In abstract, line no- 23, pituitary or pituitary gland? confirm it.

-Care should be taken between the spaces. There is a space between number and unit. Check and correct throughout the manuscript.

-The introduction section is very short. Please write little more about baicalin and copper.

-The objective of the study should be stated clearly at the end of the introduction section. Modify it by deleting the term “could affect”.

-How you chose the grouping criteria for these experiments. Did you perform any pre-experiments?

-The authors stated that baicalin-copper complex decreased the level of inflammatory cytokines, such as IFN-γ, IL-2, IL-6, IL-1β, and TNFα in challenged piglets. Did you check out the effect on any individual cell population in vitro?

-In discussion section, some references are not updated. References can be updated by citing recent references such as for baicalin: 10.1186/s13567-019-0703-6, 10.3382/ps/pez406, 10.2147/IDR.S231908 etc.

-The overall writing of the manuscript is Okay. However, abbreviations should be explained at its first use in the text. See the journal guideline about this. If needed please revise it in abstract as abbreviations are not stated in abstract.

-Why the authors select Hypothalamus and Pituitary Genes? Give reason for it in discussion section.

-I will appreciate your findings, if you draw a conclusion in the real-world prospect. How you correlate your findings and the dose of BCU to pig industry? Is it feasible for the farmers to use this dose every day as a feed supplement? Please comment on this.

-How the authors confirm that piglets received the dose evenly?

-Discuss some future directions that the authors withdraw from this study in discussion section.

Author Response

In abstract, line no- 23, pituitary or pituitary gland? confirm it.

Response:it should be pituitary

-Care should be taken between the spaces. There is a space between number and unit. Check and correct throughout the manuscript.

Response:We have added a space between number and unit in the article.

The introduction section is very short. Please write little more about baicalin and copper.

Response:We reviewed the latest research on baicalin and added it to the introduction.

The objective of the study should be stated clearly at the end of the introduction section. Modify it by deleting the term “could affect”.

Response:we have deleted ‘could’

How you chose the grouping criteria for these experiments. Did you perform any pre-experiments?

Response: We determined the dose of BCU through preliminary experiments.

The authors stated that baicalin-copper complex decreased the level of inflammatory cytokines, such as IFN-γ, IL-2, IL-6, IL-1β, and TNFα in challenged piglets. Did you check out the effect on any individual cell population in vitro?

Response: No, all the results in this article are derived from animal experiments. Cell experiment is our future research plan

In the discussion section, some references are not updated. References can be updated by citing recent references such as for baicalin: 10.1186/s13567-019-0703-6, 10.3382/ps/pez406, 10.2147/IDR.S231908 etc.

Response: Thanks for your advice, we have cited this article.

The overall writing of the manuscript is Okay. However, abbreviations should be explained at its first use in the text. See the journal guideline about this. If needed please revise it in the abstract as abbreviations are not stated in abstract.

Response: Thanks for your advice, we have checked all abbreviation in the article.

Why the authors select Hypothalamus and Pituitary Genes? Give the reason for it in the discussion section.

We analyzed a series of related hormones that DON may affect according to the previous protocol in the laboratory, but we only selected part of the results for display

I will appreciate your findings, if you draw a conclusion in the real-world prospect. How you correlate your findings and the dose of BCU to pig industry? Is it feasible for the farmers to use this dose every day as a feed supplement? Please comment on this.

This synthetic product can be applied to production. Our scientific research experiment is only a preliminary assessment, it provides data support for production, and does not accurately calculate the production cost. Therefore, the possibility of application production should be considered based on the production cost.

How the authors confirm that piglets received the dose evenly?

The feed has been thoroughly mixed so that the concentration of DON/bcu in the feed is consistent, so the DON/BCU consumed by the piglets is evenly

-Discuss some future directions that the authors withdraw from this study in discussion section

This article shows that gut microbiota is associated with inflammation and hormone secretion,we will use metagenomics, culture omics, sterile animal models and other methods for further research

Reviewer 3 Report

  • Line 9: Indicate the abbreviation of baicalin-copper complex (BCU).
  • Line 10: Avoid starting a sentence with a number.
  • Line 19 and 20: What is SS?.
  • Line 25: hormones ,
  • Line 50: Remove the bold from the term Furthermore.
  • Add a hypothesis in the introduction.
  • Provide more detailed information on the contamination of diets with DON.
  • Specify the origin of BCU, is it a commercial preparation? or was developed for this study. In this case, indicate how it was made. Indicate what concentrations of the compounds it contains.
  • Add a table with the diets used and with the composition analysis.
  • Line 272: died??.
  • It is not possible for a scientific article not to indicate the protocols of its methods. It is necessary to describe in detail the analysis of the microbiota. Methods must be reproducible.
  • Many formatting errors in the references.
  • Figure 1 and 2 are not readable, so understanding these results is complex.
  • It is recommended to select the main graphics that explain the results for a better understanding.
  • Figure 3 is not self-explanatory, the effect of the treatments is not understood. Compare the treatments.
  • The paper has a lot of information, it is necessary to synthesize and highlight the most relevant aspects. For example, in Figure 4 only the graphs that show significant differences are shown.
  • Figure 5 is not readable.
  • The results are not clear and the discussion needs to be improved.
  • Several references are not in the format of the journal.

Author Response

Line 9: Indicate the abbreviation of baicalin-copper complex (BCU).

Thanks for your advice. We have indicated the abbreviation of baicalin-copper complex.

Line 10: Avoid starting a sentence with a number.

Thanks for your advice. We have changed the sentence as you required.

Line 19 and 20: What is SS?.

Thanks for your advice. SS means Somatostatin.

Line 25: hormones ,

Thanks for your advice. We have adjusted the spaces

Line 50: Remove the bold from the term Furthermore.

Thanks for your advice. We have removed the bold from the term futhermoer.

Add a hypothesis in the introduction.

Thanks for your advice. We have modified the introduction according to your advice

Provide more detailed information on the contamination of diets with DON.

We have added one sentence about DON contamination (line 51-52)

Specify the origin of BCU, is it a commercial preparation? or was developed for this study. In this case, indicate how it was made. Indicate what concentrations of the compounds it contains.

BCU was developed for this study, and we added the BCU synthesis method in the method section

Line 272: died??.

According to the experimental process, we speculate that the piglets was died for serious weaning stress。

It is not possible for a scientific article not to indicate the protocols of its methods. It is necessary to describe in detail the analysis of the microbiota. Methods must be reproducible.

In scientific articles, method is an indispensable part. The analysis of microorganisms in this article was done by the Institute of Subtropical Sciences of the Chinese Academy of Sciences and Beijing Novogene Co., Ltd. Following your suggestions, we detailed the software and websites involved in the analysis of microorganisms in the method section, and carried out the process of data analysis

Many formatting errors in the references.

Thank you for your suggestion, we have corrected the errors in the references, , please verify that whether it meets the requirements.

Figure 1 and 2 are not readable, so understanding these results is complex.

We have changed the format of figure 1 and figure 2. But I don’t know if it meets the requirements, we will upload a pdf image in the system.

It is recommended to select the main graphics that explain the results for a better understanding.

Thanks for your advices. We put part of the picture in Figure 4 into the supplementary material.

Figure 3 is not self-explanatory, the effect of the treatments is not understood. Compare the treatments.

Thanks for your advices. Figure 3 shows the differential flora between each group. Figure A shows the differential flora between the CON and DON groups, Figure B shows the differential flora between the DON and DBCU groups, and Figure C shows the BCU and Differential flora of Con group

The paper has a lot of information, it is necessary to synthesize and highlight the most relevant aspects. For example, in Figure 4 only the graphs that show significant differences are shown.

Thanks for your advices. We put part of the picture in Figure 4 into the supplementary material.

Figure 5 is not readable.

We have modified figure 5 in tiff format. But I don’t know if it meets the requirements, we will upload a pdf image in the system.

The results are not clear and the discussion needs to be improved.

We have modified the results of the article, please verify that whether it meets the requirements.

Several references are not in the format of the journal.

Thank you for your suggestion, we have corrected the errors in the references, , please verify that whether it meets the requirements.

Reviewer 4 Report

This manuscript aims to evaluate how Baisalin-Copper complex modulates multiple responses in pigs in the presence and absence of DON. The experimental model was good and the data collected appears to be good, however, the manuscript lacked any meaningful or clear discussion of the results. Additionally, the English quality was very low and where it was not riddled with grammatical errors or sentences that were very unclear the tone was overly conversational. Specific comments that need to be addressed are below:

Throughout the manuscript Authors frequently refer to the DBCU group as BCU in Don challenged piglets. This is very confusing when reading as there is a BCU group, I recommend Authors always use the DBCU abbreviation when referring to this group, and only use BCU when referring to the BCU group.

Specific comments are below

Results:

Lines 137: Authors state that immunoglobulin was monitored during the experiment, which indicates they were evaluated at multiple time points, which well that would have been very interesting was not done in this experiment. Authors should specify these levels were checked at termination or on day 15

Discussion:

The majority of the discussion is simply restating results and has an overly casual conversation writing style that is not appropriate for the discussion of a manuscript. There is no tying together of the results from each output and any meaningful discussion of what these results mean in relation to the body of work regarding DON contaminated feed. I recommend the Authors rewrite the discussion not based on what each figure showed and rehash the results, but take the results a step further and interpret them against the body of literature as well as tying their results together.

Line 231-232: When discussing the results, the authors fail to discuss why when DON and BCU are combined they do not have an additive effect, but reduce the effect on the lactobacillus and clostridium abundance.

Line 239: Authors should eliminate the use of the word “obviously”

Line 240-242: This is simply restating the results with no discussion whatsoever.

Line 254-257: eliminate the use of “on the one hand” and “on the other hand”

Line 265-266: The state DON suppressed expression of NPY, however, they previously showed an increased serum level of NPY in DON treated animals, this should be discussed.

Line 274-276: after reading the discussion I can no clearly see how the author was able to come to this conclusion that supplementation with BCU may alleviate inflammatory responses.

Materials and method:

Was the level of DON in the control feed confirmed to be 0mg/kg DON?

Line 342: Through what method was total RNA extracted? Was there any validation of RNA integrity through either bioanalyzer or denaturing agarose gel?

More then one reference gene should be checked and if possible multiple reference genes should be used to normalize target gene expression.

The forward primer provided for GAPDH does not amplify GAPDH, I recommend authors double-check this sequence or design new primers if this is not a typo.

Author Response

Throughout the manuscript Authors frequently refer to the DBCU group as BCU in Don challenged piglets. This is very confusing when reading as there is a BCU group, I recommend Authors always use the DBCU abbreviation when referring to this group, and only use BCU when referring to the BCU group.

Thank you for your suggestion, we have modified the statement in the article according to your requirements

Lines 137: Authors state that immunoglobulin was monitored during the experiment, which indicates they were evaluated at multiple time points, which well that would have been very interesting was not done in this experiment. Authors should specify these levels were checked at termination or on day 15

We collected blood for immunoglobulin testing on the 15th day. 

Discussion:

The majority of the discussion is simply restating results and has an overly casual conversation writing style that is not appropriate for the discussion of a manuscript. There is no tying together of the results from each output and any meaningful discussion of what these results mean in relation to the body of work regarding DON contaminated feed. I recommend the Authors rewrite the discussion not based on what each figure showed and rehash the results, but take the results a step further and interpret them against the body of literature as well as tying their results together.

We have modified the discussion of the article, please verify that whether it meets the requirements.

Line 231-232: When discussing the results, the authors fail to discuss why when DON and BCU are combined they do not have an additive effect, but reduce the effect on the lactobacillus and clostridium abundance.

Thank you for your suggestion. To be honest, DON and BCU are two completely different substances. Although they can also affect the composition of intestinal flora, their pathways of action may be completely different. Therefore, they are different from each other. It is understandable that the effect does not have an additive effect. And from the text, they may have more antagonistic effects.

Line 239: Authors should eliminate the use of the word “obviously”

We have eliminate the use of the word “obviously”

Line 240-242: This is simply restating the results with no discussion whatsoever.

We have modified the discussion of the article, please verify that whether it meets the requirements.

Line 254-257: eliminate the use of “on the one hand” and “on the other hand”

We have eliminated the use of “on the one hand ”and “on the other hand ”in this article.

Line 265-266: The state DON suppressed expression of NPY, however, they previously showed an increased serum level of NPY in DON treated animals, this should be discussed.

Line 274-276: after reading the discussion I can no clearly see how the author was able to come to this conclusion that supplementation with BCU may alleviate inflammatory responses.

Materials and method:

Was the level of DON in the control feed confirmed to be 0mg/kg DON?

The DON concentration in the control diet was 124 µg/kg.

Line 342: Through what method was total RNA extracted? Was there any validation of RNA integrity through either bioanalyzer or denaturing agarose gel?

We extracted total RNA using the trizol method and verified it with enaturing agarose gel, and RNA is not degraded.

More then one reference gene should be checked and if possible multiple reference genes should be used to normalize target gene expression.

In fact, we used two housekeeping genes for gene expression correction, but we only showed one gene in the article

The forward primer provided for GAPDH does not amplify GAPDH, I recommend authors double-check this sequence or design new primers if this is not a typo.

Thank you for your suggestion. We used blast to verify the primers and found that the primers can amplify the GAPDH sequence.

Round 2

Reviewer 4 Report

The revisions have significantly improved the quality of the manuscript.